# Diagnosis and Treatment of Early Chronic Obstructive Lung Disease (COPD)

**DOI:** 10.3390/jcm9113426

**Published:** 2020-10-26

**Authors:** Joon Young Choi, Chin Kook Rhee

**Affiliations:** 1Division of Pulmonary, Allergy and Critical Care Medicine, Department of Internal Medicine, Incheon St. Mary’s Hospital, College of Medicine, The Catholic University of Korea, Seoul 06591, Korea; tawoe@naver.com; 2Division of Pulmonary, Allergy and Critical Care Medicine, Department of Internal Medicine, Seoul St. Mary’s Hospital, College of Medicine, The Catholic University of Korea, Seoul 06591, Korea

**Keywords:** COPD, early COPD, diagnostics, therapy

## Abstract

Chronic obstructive lung disease (COPD) is responsible for substantial rates of mortality and economic burden, and is one of the most important public-health concerns. As the disease characteristics include irreversible airway obstruction and progressive lung function decline, there has been a great deal of interest in detection at the early stages of COPD during the “at risk” or undiagnosed preclinical stage to prevent the disease from progressing to the overt stage. Previous studies have used various definitions of early COPD, and the term mild COPD has also often been used. There has been a great deal of recent effort to establish a definition of early COPD, but comprehensive evaluation is still required, including identification of risk factors, various physiological and radiological tests, and clinical manifestations for diagnosis of early COPD, considering the heterogeneity of the disease. The treatment of early COPD should be considered from the perspective of prevention of disease progression and management of clinical deterioration. There has been a lack of studies on this topic as the definition of early COPD has been proposed only recently, and therefore further clinical studies are needed.

## 1. Introduction

Chronic obstructive pulmonary disease (COPD) is a major public-health issue affecting 328 million people [1] and is the third leading cause of mortality worldwide [2]. In addition, it represents a considerable economic burden, costing USD 18 billion annually in the USA in hospitalization costs associated with exacerbation of the disease [3].

The major pathophysiology of COPD is irreversible obstruction of the airway with progressive lung function decline, especially in patients with continuous exposure to risk factors such as cigarette smoke, biomass smoke exposure, and air pollution, although lung-function trajectories have a milder course in some patients [4,5,6]. Therefore, it is important to halt the lung-function decline in patients who do not meet the diagnostic criteria for COPD but are at risk of progression (pre-COPD) to overt COPD [7,8,9,10]. In addition, early diagnosis of preclinical COPD may be crucial in patients with fixed obstruction on spirometry with no or only mild symptoms before they develop clinically important deterioration, such as decline in lung function, worsening of symptoms, or acute exacerbation [11,12,13]. To avoid overtreatment of pre-COPD or preclinical COPD patients, it is important to understand the diverse lung-function trajectories of COPD patients and classify those who may not eventually progress to overt COPD or severe COPD [6,14,15,16,17].

This review discusses early COPD, including variable lung-function trajectories, to understand its clinical course, provide a definition, and develop diagnostic tools that may identify COPD in the early clinical course, together with treatments that may prevent disease progression and clinical deterioration.

## 2. Variable Lung-Function Trajectories

Several recent studies have indicated that the lung-function trajectories of patients with COPD vary [6,14,15,16,17,18,19]. Genetic predisposition, sex, ethnicity, and hormonal susceptibility may all be in play, but the most important factors are early life disadvantages and risk-factor exposure, especially smoking tobacco (Figure 1) [20,21,22,23,24,25,26,27,28]. As shown in Figure 1, early life disadvantages mainly specify starting points for the decline in a patient’s lung function and represent the y-intercept of the trajectory curve [26]. On the other hand, risk factor exposure, represented by smoking tobacco, determines the gradient of the slope [23]. 

Early life disadvantages represent impediments to normal lung growth or alveolar formation in the early stages of life from the fetal stage to adolescence. The development of bronchi starts as lung bud initiation at gestational days 21–28, and continues with dichotomous and lateral branching of lung epithelium to achieve the final shape of the bronchial tree at gestational weeks 3–17 (embryonic, pseudoglandular stage) [29]. Bronchial epithelial cells differentiate into ciliated, nonciliated, and secretory cells at gestational weeks 16–26 (canalicular stage), followed by the saccular stage, when branching of terminal acinar tubules and expansion of air space size occurs from gestational week 24 to birth. Finally, true alveoli develop from terminal saccules from gestational week 36 until 18 months postnatally. Lung function and lung volume increase as the thorax grows, reaching a peak at around age 20 years [16]. 

Among these processes, there are a number of early life disadvantages that may impede the development of the lung, resulting in decreased lung function at an early age. Maternal smoking during pregnancy [30,31,32], low birthweight [33], and preterm delivery [32,34,35] have been reported to be important risk factors for impaired lung function. From birth to adolescence, smoking at a young age or passive smoking [18,24,26,36], asthma [26,37,38,39], history of respiratory infection [26,40], and chronic bronchitis [41,42] are associated with increased risk of lung-function impairment. These early life disadvantages interfere with the achievement of sufficient peak lung function, which is usually reached at around age 20 years [40,43,44]. From the early 20s to mid-30s, lung function tends to reach a plateau or continues to increase slowly [45,46]. Exposure to tobacco smoke shortens this period and may result in early decline in lung function [45].

Smoking is not only associated with the early life disadvantages discussed above but also facilitates lung-function decline at all ages, as shown by the steeper gradient of the lung-function trajectory curve in Figure 1. The magnitude of the role that smoking plays in reducing lung function differs between individuals, which may be associated with various factors, such as genetic susceptibility [47,48], sex [49,50,51,52], and ethnicity [52]. As a result, only a minority of smokers are consequently diagnosed with COPD in their life course [53]. In addition, quitting smoking changes the steeper lung trajectories of smokers to a gradient similar to that of non-smokers [23]. Risk factors other than smoking, such as biomass smoke exposure, air pollution, and occupational exposures, may also be associated with decreased lung function, but there have been insufficient studies on patterns of lung-function decline due to other risk factors or changes in lung function following discontinuation of the exposure [54,55,56,57]. In addition, exacerbation events of airway diseases cause a temporarily substantial decrease in lung function, and after stabilization this decrease usually does not recover fully to the pre-event lung function [58,59]. The impact of the exacerbation rate on forced expiratory volume in 1 s (FEV1) decline is more profound in early COPD [58]. 

## 3. Diagnosis

### 3.1. Definition

The concept of early COPD should be understood from the perspective of the longitudinal course of the disease. This represents an earlier point in the course of COPD that does not yet show spirometric airway obstruction or typical clinical manifestations [16]. It should be distinguished from “mild COPD,” which is generally perceived as a cross-sectional-perspective mild spirometric airway obstruction regardless of the point in the course of the disease [57,60,61,62,63]. However, due to a lack of evidence to identify COPD patients in the early stages of the course of the disease, many groups have used the definition of mild COPD without distinguishing it from early COPD.

Recently, Martinez et al. [15] proposed a definition of early COPD in which diagnosis is made in patients <50 years of age with a smoking history ≥ 10 pack-years and fulfilling one or more of the following: FEV1/forced vital capacity (FVC) below the lower normal limit; compatible computed tomography (CT) abnormalities, such as visual emphysema, air trapping, or bronchial thickening; evidence of accelerated FEV1 decline (≥60 mL/year). This definition was established based on operational and practical considerations assuming that patients at an early age may not have developed significant COPD. In addition, this definition included only patients with ≥10 pack-years of cigarette smoking, which is regarded as the minimum exposure required for the pathogenesis of COPD. While this definition can be useful for enrollment of patients in academic studies, in clinical fields it may have some limitations. Moreover, this definition focuses only on cigarette smoking, the main risk factor of COPD, and neglects patients with other important risk factors, such as biomass smoke exposure, air pollution exposure, previous infection, or occupational exposure [64]. These types of non-smoking-related COPD account for 25–45% of all COPD cases, and this definition has the limitation that it neglects this important phenotype in non-smokers [65]. Despite these limitations, this definition is valuable in that it may allow the selection of early COPD patients in large cohort studies or allow inclusion of patients in large randomized controlled studies regardless of their current lung-function status, in contrast to studies regarding mild COPD.

As the definition of early COPD states that airway obstruction is not necessarily a requirement, early COPD can be divided into two subtypes, i.e., pre-COPD and preclinical COPD (Figure 2). Pre-COPD represents patients with no airway obstruction (FEV1/FVC ≥ 0.7) who are at risk of progression to COPD. These patients may have early pathological changes in their lungs or accelerated lung-function decline without reaching the cutoff value for COPD [66,67]. Destruction of small conducting airways <2 mm in diameter (terminal bronchioles) is one of the most important features in the early phase of progression from the normal lung to early COPD, which precedes emphysematous destruction [67,68]. The selection of patients at risk and modification of risk factors are keys to avoiding progression to overt COPD in this group of incipient patients.

On the other hand, preclinical COPD represents patients with airway obstruction (FEV1/FVC < 0.7) but no/mild respiratory symptoms, such as cough, sputum, or dyspnea. The cutoff COPD Assessment Test (CAT) score to identify this group may be derived from the value used in Global Initiative for Chronic Obstructive Lung Disease (GOLD) staging (<10; less-symptoms group, ≥10; more-symptoms group) [69]. A previous study showed that 13.7% of smoking-related COPD patients do not have symptoms, and 1.5% of asymptomatic smokers have airway obstruction [70]. These patients are likely to be underdiagnosed, exposed continuously to risk factors, especially tobacco smoking, and progress to overt COPD. Early detection of preclinical COPD may allow prevention of progression to the advanced stages of the disease course by removing risk factors, avoid clinically important deterioration, such as exacerbation or pneumonia, and manage associated comorbidities [12,71]. 

### 3.2. Diagnostic Tools for Early COPD

As the definition of early COPD was established based on operational and practical considerations and has several limitations, the diagnosis of early COPD must be made by comprehensive measurements in the clinical field. Clinicians should examine exposure to all risk factors related to various lung-function trajectories affected by early life disadvantages and risk factor exposures from utero (Table 1). In addition, several studies have attempted to detect early changes in the airway and lung tissue indirectly using various physiological, radiological, and laboratory tests.

#### 3.2.1. Identification of Risk Factors

Smoking is the most important risk factor for COPD. The harmful effects of tobacco smoking on lung function begin from the fetal stage with maternal exposure. A study of 8863 non-smoking children aged 8–12 years who were exposed to maternal smoking showed a reduced expiratory flow volume on spirometry [72]. In addition, a longitudinal prospective study with a 21-year follow-up showed a significant association between maternal smoking during pregnancy and reduction in FEV1 in male patients [73]. Exposure to second-hand smoke in the childhood to the young-adult stages, mainly from parents, is also an important risk factor for adult obstructive lung diseases [36,74]. Finally, tobacco smoking is the most important risk factor for COPD [75], and this is well established from adolescence to old age [23,24].

Risk factors other than cigarette smoking, including childhood infection or respiratory diseases such as asthma, biomass smoke exposure, air pollution exposure, and occupational exposures, are also important but have often been overlooked in the assessment of COPD patients. Non-smoking-related COPD accounts for 25–45% of COPD cases, which is much higher than clinicians usually expect [76].

Past histories of childhood respiratory infection or asthma are also important risk factors. Studies have shown that respiratory infection in childhood is associated with chronic-bronchitis symptoms, decreased lung function, increased airway wall thickness, and increased COPD exacerbation in adulthood [33,77,78]. Hayden et al. proposed two possible explanations for these risk factors—childhood infections may cause changes in the airway that result in lung diseases in adulthood, or underlying lung conditions in childhood may increase the risk of childhood pneumonia and future lung diseases [77].

Most large-population COPD studies or randomized controlled studies have excluded patients with asthma due to the difference in pathophysiology between the two diseases. Inclusion of asthma patients in the COPD arm may introduce serious bias, as the pathophysiology of asthma is mainly chronic allergic airway inflammation with airway variability. However, a 15-year follow-up of lung function in asthma patients showed a substantial decrease in FEV1 compared to lung function in patients without asthma [37]. In addition, previous studies have shown that patients with a history of early or late asthma have a 10–20-fold increase in risk of airway obstruction, and these patients may progress to early COPD [38,79]. After smoking, airway hyperresponsiveness is the most important risk factor for COPD, which is responsible for 15–17% of new cases in young adults [80]. Patients with a history of asthma who develop COPD in later years may be categorized as asthma–COPD overlap (ACO) according to the ATS roundtable criteria and modified Spanish COPD guidelines [81]. These patients are more symptomatic, experience more exacerbations, and have higher hospitalization rates [82,83]. 

One of the most important non-tobacco risk factors may be the burning of biomass fuel, to which about 3 billion people are exposed worldwide, with the proportion being higher in developing countries, where it may affect more than 80% of the population [76]. This exposure rate is almost three times higher than that of cigarette smoking, and as the rate of biomass fuel usage is higher in developing countries, COPD due to biomass smoke exposure may have been underestimated. These biomass smoke exposure-related COPD patients are more likely to be female with better lung function, but similar symptoms, exercise capacity, and quality of life (QOL) compared to patients with smoking-related COPD [84]. In addition, previous studies have shown that biomass smoke exposure-related COPD predominantly involves airways with less emphysematous changes in the lung compared to smoking-related COPD [65,85,86]. 

Occupational exposures are also important risk factors that are usually unappreciated or neglected. The estimated population attributable risk of occupational exposure for COPD is reported to be 12–55% [87]. These occupational exposures include exposure to dust (coal mining, hard-rock mining, concrete manufacturing, construction, tunneling, brick manufacturing, iron and steel founding, gold mining), animal farming, crop farming, chemicals (plastics, textiles, rubber industry, leather manufacturing), diesel exhaust, and road dust (sweeping) [55]. As occupational exposures are not well appreciated as causes of COPD in the clinical field, physicians should be more aware of these types of exposure, especially in cases of non-smoking-related COPD.

Furthermore, outdoor air pollution has been shown to be an important risk factor for the prevalence and exacerbation of respiratory diseases. Air pollution consists mainly of pollutants emitted from vehicles and factories. Studies have shown that the effects of air pollution are found more in women than in men. A cross-sectional cohort study demonstrated an association between traffic exposure and poor lung function, which was only statistically significant in women [88]. A meta-analysis of 6550 subjects showed that NO_2_, NO_x_, PM_10_, and traffic indicators were significantly associated with COPD only in female patients [89]. A 30-year longitudinal Canadian study also demonstrated an association between increases in fine particulate matter (PM_2.5_) and prevalence of COPD in women (OR = 1.57) [90]. On the other hand, it has also been suggested that air pollution has harmful effects on lung development in 10–18-year-old adolescents, which may be regarded as an early life disadvantage in the lung-function trajectory [91]. 

Finally, genetic factors should be taken into consideration in patients with suspected early COPD. Not only severe α1-antitrypsin (AAT) deficiency, which is a well-established genetic syndrome, but also cutis laxa, Marfan syndrome, and Ehlers-Danlos syndrome may be associated with COPD, as these syndromes present with emphysema, lung blebs, and pneumothorax [92]. In addition, some variations in genetic loci, such as *CHRNA3/5*, *HHIP*, and *FAM13A*, are associated with a genetic risk of COPD [25]. Further investigations are needed for the clinical application of these observations.

#### 3.2.2. Pulmonary Function Test and Other Physiological Tests

The pulmonary function test (PFT) is the most important physiological test in COPD patients, as it confirms airway obstruction and assesses the severity of airflow limitation [69]. Evidence of accelerated FEV1 decline >60 mL/year is one of the selective conditions for diagnosis of early COPD in the definition proposed by Martinez et al. [15]. The threshold of 60 mL/year was determined roughly by doubling the normal FEV1 decline in non-smokers, which is 25–30 mL/year. However, Washko et al. analyzed data from the longitudinal CARDINA cohort study to categorize lung-function trajectories in young adults, and showed a mean FEV1 decline of 51 mL/year in the accelerated lung-function decline group [17]. Lange et al. also analyzed data from three large cohort studies to investigate lung-function trajectories. They reported a mean FEV1 decline of 53 mL/year in those who initially had no airway obstruction and later developed COPD after 22 years of obstruction [17]. These results indicate that the threshold of 60 mL/year may omit some individuals with rapid decline who are at risk of future COPD. Nevertheless, the cutoff level of FEV1 for individuals with rapid decline has not been fully validated, and longitudinal and sequential measurement of FEV1 remain essential.

In addition to FEV1, several elements of the spirometry test may also predict patients who are in the early phase of COPD. Although forced expiratory flow between 25% and 75% of vital capacity (FEF_25–75_) is associated with small airway obstruction, this measure has seen only limited use due to its high variability between subjects. However, as early changes in the pathogenesis of COPD are mainly in the small conducting airways, FEF_25–75_ should be reconsidered in early detection of COPD. A recent study based on the Isle of Wight Birth Cohort investigated longitudinal lung-function changes from birth to 26 years [93]. Early lung-function changes resulting from smoking were significant in FEF_25–75_ and FEV1/FVC but not in FEV1. In addition, airway hyperresponsiveness has also been shown to be an important risk factor associated with a fourfold greater risk of overt COPD [80]. As it is a key pathophysiological feature of asthma, the presence of airway hyperresponsiveness in COPD patients may indicate the coexistence of asthma, which is an important risk factor for COPD. However, airway hyperresponsiveness in COPD patients does not always have asthmatic features [94]. Various factors, including increased airway smooth muscle cell contractility or airway thickness, reduced elastic recoil, and inflammatory/neurogenic consequences mediated by smoking, may be associated with airway hyperresponsiveness in COPD.

Lung hyperinflation and air trapping are important features of COPD that appear not only in severe cases but also in mild COPD [95]. Studies have shown that the ratio of residual volume to total lung capacity (RV/TLC) can be used in the diagnosis and assessment of early COPD [96,97]. Elevated RV/TLC may be present in pre-COPD and preclinical COPD, and predict poor lung function and a better bronchodilator response. Radiologically measured RV/TLC has also been studied, as will be discussed in a later section. In addition, lung hyperinflation may become more prominent as the respiratory rate rises in response to exercise, resulting in exertional dyspnea. This corresponds with the results of a previous study indicating that preclinical COPD patients have poor exercise tolerance compared to normal controls, as assessed by cardiopulmonary exercise tests [98].

The diffusion capacity of the lung for carbon monoxide (DL_CO_), a parameter associated with alveolar destruction, is reduced in some smokers without airway obstruction [99]. In addition, smokers with normal lung function showed >7-fold increased risk of the development of COPD in a decreased DL_CO_ group compared with a normal DL_CO_ group [100]. This indicates that decreased DL_CO_ may predict later development of COPD.

The potential roles of other physiological indices for early detection of COPD have also been studied. The lung-clearance index is associated with the heterogeneity of small airway function, which has been shown to be associated with future development of airway obstruction and COPD in males [101]. Impedance oscillometry, which measures airway resistance and capacitance, may also have a role in detecting early COPD [102,103,104]. 

#### 3.2.3. Imaging Studies

Emphysematous changes due to the destruction of alveoli below the terminal bronchioles and small-airway disease caused by airway inflammation and remodeling are important changes that occur in the pathophysiology of COPD. The most important radiological method for evaluating these changes is chest-computed tomography (CT). CT is essential for distinguishing structural abnormalities associated with COPD. The majority of patients with emphysema were reported to be at GOLD stage 0, which may be classified as pre-COPD [105]. For these patients, it is necessary to assess the possibility of further progression to overt COPD. Several studies have shown that if emphysema is more than moderate in severity or is 10% on quantitative analysis, patients may show a rapid decline in FEV1 in their lifetime trajectory [106,107,108]. Longitudinal lung-function follow-up may be needed in selected patients with emphysema at risk of further development of COPD. 

Another significant pathological change in COPD is disease in small airways of <2 mm in diameter. As 40% of terminal bronchioles are lost in mild COPD patients, assessment of small-airway disease may be crucial in detecting early COPD [68]. Assessment of small-airway disease in vivo may be challenging because the airways are too small for visualization using the current resolution of CT. To overcome these limitations, indirect measurements of functional small-airway disease have been studied using inspiratory and expiratory chest CT [109]. Parametric Response Mapping (PRM) is a novel technique that uses software to identify voxels of emphysema area (PRM^emph^) as HU < −950 on inspiration and HU < −856 on expiration, and small-airway disease (PRM^fSAD^) as HU > −950 on inspiration but HU > −856 on expiration [110]. In a 5-year longitudinal analysis, the COPDGene study showed that only PRM^fSAD^ was associated with FEV1 decline rates in GOLD 0 patients, while both PRM^fSAD^ and PRM^emph^ were associated with FEV1 decline rates in GOLD 1–4 patients [111]. In addition, in GOLD 1–4 patients, the proportional contribution of PRM^fSAD^ to FEV1 decline was significant in GOLD 1–2 patients compared to more severe groups. These results imply that PRM^fSAD^ may be an important indicator of early COPD.

The total airway count (TAC) decreases with the progression of small-airway disease due to narrowing and destruction of terminal bronchioles. TAC determined by CT using image analysis software has been shown to be associated with FEV1, FEV1/FVC, and rates of FEV1 decline [112]. Airway wall thickness has also been measured in chest CT in patients regardless of airflow limitation, and has also been shown to be associated with FEV1 decline rate [113]. These studies may help to identify patients at risk of rapid FEV1 decline. Two recent studies assessed lung volume based on inspiratory and expiratory CT scans [114,115]. The results showed that high radiographically measured RV/TLC was associated with rapid decline in FEV1/FVC in smokers with normal spirometry, which may eventually progress to overt COPD. 

Other imaging modalities have been studied in the assessment of early COPD. Hyperpolarized magnetic resonance imaging (MRI) provides additional information on structural and functional abnormalities of the lung, regional ventilation, alveolar enlargement, and gas diffusion [116]. This technique has potential benefits for detection of patients at risk of COPD [116,117]. In addition, gadolinium-enhanced MRI perfusion scans may play a role in detection of early structural changes of COPD, but further studies are needed for validation [118,119].

#### 3.2.4. Clinical Features 

Chronic bronchitis is one of the most important phenotypes of COPD, which is associated with poor health-related QOL, poor lung function, frequent exacerbation, and is even associated with a higher mortality rate [120,121,122,123]. It is usually defined by frequent cough and sputum production for 3 months per year for over 2 consecutive years, but definitions using sub-questionnaires of St. George’s Respiratory Questionnaire (SGRQ) scores or CAT scores have also been validated [124,125,126,127]. Chronic bronchitis may develop with or without airway obstruction [10,128,129]. Among patients without airway obstruction, previous studies have shown that the presence of chronic bronchitis symptoms is associated with poor lung function, increased risk of exacerbation, and rapid FEV1 decline, which may result in future COPD [130,131,132]. 

The SGRQ is a well-validated questionnaire that is associated with health-related QOL. In post hoc analysis of the UPLIFT study, COPD patients < 50 years old had lower SGRQ scores compared to the entire age group [133]. Therefore, increased SGRQ scores may help to detect undiagnosed young COPD patients. In addition, based on a previous study showing impaired exercise capacity in preclinical COPD patients, symptoms of exertional dyspnea or impaired exercise function assessed by the 6-min walking test (6MWT) may be helpful in early detection of COPD [98]. Further validation of these clinical scores is needed. 

#### 3.2.5. Biomarkers of Disease Progression 

Various biomarkers have been studied but still need to be validated. A recent prospective study in China showed that inflammatory markers, including interleukin (IL)-6, high-sensitivity C-reactive protein (hs-CRP), and microRNA-23a (miR-23a) in peripheral blood and pH values of exhaled breath condensate are associated with rapid decline of FEV1 in non-COPD patients [134]. In addition, analysis of serum biomarkers showed that Club Cell Secretory Protein (CC16), soluble receptor for advanced glycation end products (sRAGE), and fibrinogen were associated with FEV1 decline in both COPDGene and ECLIPSE cohorts [135]. Proteomics analysis of mild to moderate COPD patients showed that proteins associated with inflammation, blood coagulation, complement pathways, oxidative stress, and lipid metabolism may be potential biomarkers for the early detection of COPD [136]. However, there have been few reports on this subject, and further studies are needed for clinical application. 

### 3.3. Natural Course of Early COPD 

An analysis of Korean national data showed increases in healthcare utilization and total medical costs of preclinical GOLD I–II COPD patients between 2007 and 2012 [137]. This implies that some preclinical COPD patients may eventually progress to COPD and require more medical resources. In addition, asymptomatic, undiagnosed COPD was associated with more frequent exacerbation and pneumonia and a higher mortality rate compared to non-COPD patients [12]. Similar results were obtained in an analysis of a Danish cohort in the Copenhagen General Population study, which showed that COPD patients <50 years old with a history of smoking had chronic respiratory symptoms, severe airflow limitation, frequent hospitalization, and a higher mortality rate compared to the healthy general population [138]. Based on these results, clinicians should be aware of patients in the preclinical or mild stage of the disease and make timely decisions for medical interventions. In addition, as only a few patients at GOLD stage 0 develop symptomatic COPD, a means of identifying those who may eventually have airway obstruction must be developed [66].

### 3.4. Treatment of Early COPD

The prevalence of early COPD may be underestimated and has been reported to be associated with substantial symptoms, risk of exacerbation, lung-function decline, and a poor health-associated QOL [139,140,141]. Although there is increasing interest in early COPD, there have been only a few studies related to treatment in groups of patients corresponding to the recent definition of early COPD. To gain insight into the best treatment options, we may refer to previous studies with target populations sharing similar characteristics. In this section, we will discuss previous studies investigating the treatment of early COPD, as assessed using various definitions. 

Patients at risk of COPD who have not reached the cutoff for airway obstruction in lung-function tests have been classified as having pre-COPD [142]. The identification of patients who will eventually develop clinically significant COPD is important, as the majority of the smoking population show preservation of normal lung function until the end of their lives [53]. Early interventions for prevention or to impede the progression of the disease should be applied in these patients. Cessation of smoking is expected to play the greatest role in prevention or slowing the progression of the disease in both overt COPD and pre-COPD. The classic Fletcher-Peto model has shown that cessation of smoking results in recovery of the normal rate of lung-function decline, versus an accelerated rate before cessation [23]. The results of the Lung Health Study, a multicenter randomized clinical trial in COPD patients with mild-to-moderate airway obstruction, showed that an intensive smoking intervention program significantly halted FEV1 decline in middle-aged smokers [143]. A follow-up study in patients who had actually succeeded in quitting smoking correspondingly showed a favorable outcome in terms of FEV1 improvement in the first year (average of 47 mL), and halving of the rate of FEV1 decline by the 5-year follow-up compared to those who continued to smoke, regardless of smoking quantity, age, baseline lung function, or airway hyperresponsiveness [144]. Furthermore, this intensive smoking intervention resulted in significant improvement in respiratory symptoms and decreased mortality [145,146]. As the rate of FEV1 decline was decreased in quitters compared to sustained smokers regardless of initial airway obstruction, cessation of smoking may prevent the progression of pre-COPD to overt COPD [147,148]. Avoidance of other risk factors, such as indoor/outdoor pollution or occupational exposure, may also be important, as these are often overlooked but important causes of COPD development. Furthermore, influenza or pneumococcal vaccination in high-risk groups may be helpful to avoid acute exacerbation, which causes a substantial decrease in FEV1 even after recovery from the event [58,140]. Pharmacological interventions in pre-COPD patients have yet to be studied. A clinical trial in pre-COPD patients using indacaterol/glycopyrrolate versus placebo is currently underway, and will probably provide some insights for medical intervention in this group (REdefining THerapy in Early COPD for the Pulmonary Trials Cooperative (RETHINC); ClinicalTrials.gov identifier NCT02867761). 

### 3.5. Treatment of Early-Onset COPD and Mild COPD

Although the recent definition of early COPD by Martinez et al. specifies age <50 years old, there have been few studies in this early-onset COPD group. Subgroup analysis of the Understanding Potential Long Term Impact on Function with Tiotropium (UPLIFT) study in COPD patients ≤50 years old indicated that use of tiotropium improves SGRQ scores, alleviates lung-function decline, and reduces the rate of acute exacerbations [133]. However, these early-onset COPD studies, as with the limitations of the definition itself, involve COPD patients in whom the disease is already advanced at a young age, which should be taken into account when interpreting the results.

On the other hand, the term “early COPD” has been used interchangeably with the term “mild COPD” (GOLD I or I–II) in previous studies. [144,145,149,150,151]. However, as these studies included “early COPD” patients classified only according to disease severity in a cross-sectional view rather than targeting patients in the early phase of the disease from a longitudinal perspective, these studies are limited in that they likely included COPD patients in the late phase of the disease with a slowly progressing clinical course. 

Most of the positive results were derived from studies investigating the clinical effects of long-acting bronchodilators. The MISTRAL study group showed that patients receiving tiotropium in both GOLD I–II and GOLD III–IV groups showed significant reductions in annual number of exacerbations compared to the placebo control group [152]. In addition, subgroup analysis of UPLIFT in GOLD II patients demonstrated that use of tiotropium slowed the rate of FEV1 decline, increased SGRQ scores, and extended the time to first exacerbation or time to exacerbation requiring hospitalization compared to the placebo group [153,154,155]. Furthermore, in a study in GOLD I and II patients, Zhou et al. reported higher FEV1 and lower FEV1 decline rates in the tiotropium group compared to the placebo group [149]. Dual bronchodilator therapy, such as umeclidinium/vilanterol, has recently been shown to have beneficial effects on lung function across all severity stages, including GOLD stage II COPD [156].

Another regimen that has shown positive results is inhaled corticosteroid (ICS) plus long-acting β-agonist (LABA), although there have been relatively few studies of this combination. In the TORCH study, the efficacy of salmeterol plus fluticasone propionate (SFC) was examined in various GOLD severity stages [157]. Patients treated with SFC showed a reduction in moderate-to-severe exacerbation and improved SGRQ scores and FEV1 in all severity groups, including GOLD II [157]. Of note, use of SFC reduced mortality in GOLD stage II patients compared to placebo, which was not confirmed for long-acting muscarinic antagonist (LAMA) or ICS monotherapy [153,157,158]. In addition, the SUMMIT investigators studied the effects of fluticasone furoate (FF), vilanterol (VI), and their combination (FF/VI) in GOLD stage II patients and reported that patients treated with FF or FF/VI showed significant benefits with regard to the FEV1 decline rate [159]. In contrast, evidence for ICS monotherapy has been variable and it has even been reported to have harmful effects, and so further validation is required [158,160,161,162]. In addition, other agents, including short-acting muscarinic antagonists [143,145,163], theophylline [164], and *N*-acetylcysteine [165,166], did not show beneficial effects.

## 4. Conclusions

There is increasing interest in early COPD to understand the pathophysiological and clinical changes in the early phase of the disease, to detect at risk or undiagnosed COPD patients before they develop overt COPD, and to provide the most beneficial treatment for prevention of disease progression and to alleviate clinical deterioration. Previously, the definition of early COPD has varied, and it has often been based on mild airway obstruction, which is indistinguishable from mild COPD. However, to focus on the early phase of the disease, the recently proposed definition specifies patients <50 years old. Although this definition has some limitations, we expect that it will make it easier to include early COPD patients in future clinical studies. As the clinical phenotypes and disease status vary in early COPD, a comprehensive approach is needed to assess these patients. Assessment of small-airway disease is one of the most important issues in the diagnosis of early COPD, and recent studies on radiological measures are promising. There is still a paucity of studies on treatment of early COPD according to the recently defined definition, and further clinical studies are required.

## Figures and Tables

**Figure 1 jcm-09-03426-f001:**
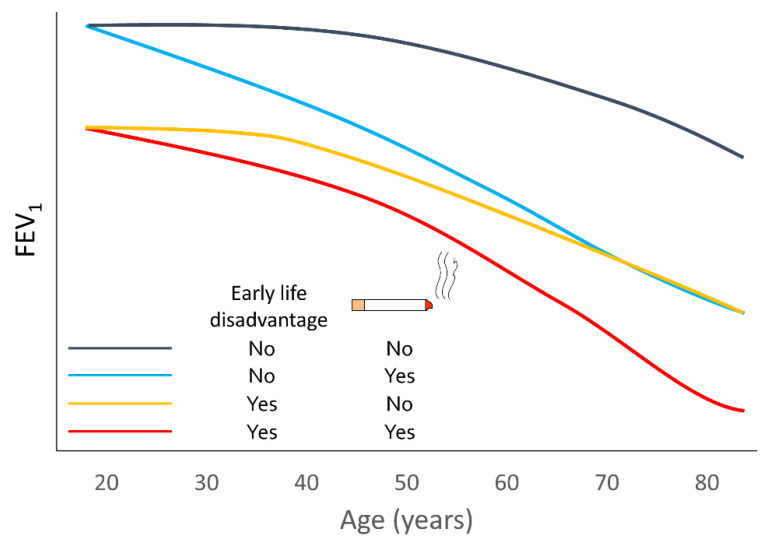
Lifelong lung function trajectories according to presence of early life disadvantages and tobacco exposures.

**Figure 2 jcm-09-03426-f002:**
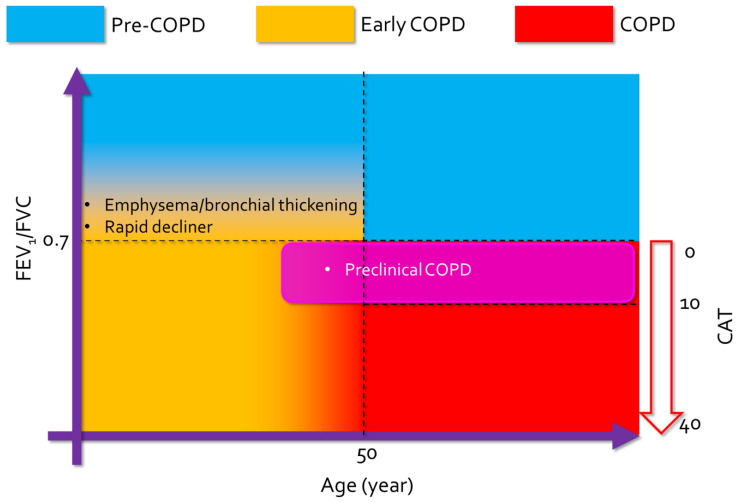
Definition of early COPD, pre-COPD and overt COPD.

**Table 1 jcm-09-03426-t001:** Diagnostic tools for early COPD.

Category	Diagnostic Tools	Supplements
Identification of risk factors	Tobacco smoking	*In utero* Parental smoking Second-hand smoke Smoking
	Childhood infection	
	Respiratory diseases (e.g., asthma)	
	Biomass smoke exposure	
	Air pollution	NO_2_, NO_x_, PM10, traffic indicators
	Occupational exposure	
	Genetic factors	AAT deficiency, cutis laxa, Marfan syndrome, Ehlers-Danlos syndrome
Physiological tests	Accelerated FEV1 decline	Annual decline >60 mL
	FEF_25–75_	Small-airway obstruction
	Airway hyperresponsiveness	
	RV/TLC	Lung hyperinflation
	Cardiopulmonary exercise test	
	DL_CO_	Alveolar destruction
	Lung clearance index	Heterogeneity of small-airway function
	Impedance oscillometry	Airway resistance and capacitance
Imaging studies	Chest CT	Distinguishing structural deformities Quantification of emphysema TAC Airway wall thickness Radiographically measured RV/TLC
	Parametric Response Mapping (PRM)	Identification of small-airway disease
	Hyperpolarized MRI	Structural and functional abnormality of lung Regional ventilation Alveolar enlargement Gas diffusion
	Gadolinium-enhanced MRI	Early structural change of COPD
Clinical features	Chronic bronchitis symptom	Cough, sputum in 3 months per year (≥2 consecutive years)
	SGRQ score	Health-related QOL
	6 MWT	Exercise function

6 MWT, 6-min walking test; QOL, quality of life; RV/TLC, ratio of residual volume to total lung capacity; SGRQ, St. George’s Respiratory Questionnaire; TAC, total airway count.

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
