# Peer review of "Diagnosis and Treatment of Early Chronic Obstructive Lung Disease (COPD)"

_jcm, 2020, doi:10.3390/jcm9113426_

Round 1
Reviewer 1 Report
Well done review. Very good.
As a minor comment:
no sentences have been proposed on the relation of "early COPD" and different COPD phenotyping ie bronchiolity vs enphyisematous
Author Response
Thank you for your valuable question. As you mentioned, there are various phenotype of COPD that may present different clinical manifestations. However, the concept of 'early COPD' has been proposed very recently, and there are lack of studies that focus on different phenotypes of early COPD. Further studies are needed on the basis of the new definition of 'early COPD', such as different lung function decline, annual exacerbation rates, overt symptoms depending on different phenotypes.
Reviewer 2 Report
The authors' review of the early COPD is complete and interesting.
I have some minor revisions to suggest concerning the two images.
Figure 1 :
- The symbols used to describe the curves "X" "O" are unclear
- FEV1 values are not reported in the vertical axis
- A more precise description in the legend of the figure is necessary
Figure 2 :
-I think it would be useful to indicate more clearly the beginning of the "CAT" arrow (Not applicable in Pre-COPD)
-It would be more useful to indicate an age scale in the horizontal axis or to eliminate the number "40"
- I would use a different color for the "Preclinical COPD" panel
- A more precise description in the legend of the figure is necessary
Author Response
Figure 1 :
- The symbols used to describe the curves "X" "O" are unclear
- FEV1 values are not reported in the vertical axis
- A more precise description in the legend of the figure is necessary
Answer)
- The symbols used to describe the curves has been changed (X -> No, O -> Yes)
- You have advised to describe exact FEV1 values in the vertical axis. However, figure 1 is a schematic graph that shows overall lung function decline according to early life disadvantages and tobacco exposures. Therefore it may be difficult to show exact FEV1 values in the vertical axis.
- We developed the figure legend as 'Lifelong lung function trajectories according to presence of early life disadvantages and tabacco exposures.'
Figure 2 :
-I think it would be useful to indicate more clearly the beginning of the "CAT" arrow (Not applicable in Pre-COPD)
-It would be more useful to indicate an age scale in the horizontal axis or to eliminate the number "40"
- I would use a different color for the "Preclinical COPD" panel
- A more precise description in the legend of the figure is necessary
Answer)
- We have changed the "CAT" arrow that shows not applicable in pre-COPD more clearly.
- We erased number "40" in the X- axis (Age).
- We changed the color of 'preclincal COPD' into purple.
- We changed the figure legend into "Definition of early COPD, pre-COPD and overt COPD'.